# Derivation of correlation dimension from spatial autocorrelation functions

Yanguang Chen *

Department of Geography, College of Urban and Environmental Sciences, Peking University, Beijing, PRC

* chenyg@pku.edu.cn

## Abstract

### Background

Spatial complexity is always associated with spatial autocorrelation. Spatial autocorrelation coefficients including Moran's index proved to be an eigenvalue of the spatial correlation matrixes. An eigenvalue represents a kind of characteristic length for quantitative analysis. However, if a spatial correlation process is based on self-organized evolution, complex structure, and the distributions without characteristic scale, the eigenvalue will be ineffective. In this case, a scaling exponent such as fractal dimension can be used to compensate for the shortcoming of characteristic length parameters such as Moran's index.

### Method

This paper is devoted to finding an intrinsic relationship between Moran's index and fractal dimension by means of spatial correlation modeling. Using relative step function as spatial contiguity function, we can convert spatial autocorrelation coefficients into spatial autocorrelation functions.

### Result

By decomposition of spatial autocorrelation functions, we can derive the relation between spatial correlation dimension and spatial autocorrelation functions. As results, a series of useful mathematical models are constructed, including the functional relation between Moran's index and fractal parameters. Correlation dimension proved to be a scaling exponent in the spatial correlation equation based on Moran's index. As for empirical analysis, the scaling exponent of spatial autocorrelation of Chinese cities is $D_c = 1.3623 \pm 0.0358$, which is equal to the spatial correlation dimension of the same urban system, $D_2$. The goodness of fit is about $R^2 = 0.9965$. This fractal parameter value suggests weak spatial autocorrelation of Chinese cities.

### Conclusion

A conclusion can be drawn that we can utilize spatial correlation dimension to make deep spatial autocorrelation analysis, and employ spatial autocorrelation functions to make complex spatial autocorrelation analysis. This study reveals the inherent association of fractal

**Data Availability Statement:** All relevant data are within the paper and its Supporting Information files.

**Funding:** The author received specific funding for this work from the National Natural Science

Foundation of China (Award Number: 42171192; Recipient: Yanguang Chen). The funders had no role in study design, data collection and analysis, decision to publish, or preparation of the manuscript.

**Competing interests:** The authors have declared that no competing interests exist.

patterns with spatial autocorrelation processes. The work may inspire new ideas for spatial modeling and exploration of complex systems such as cities.

## 1 Introduction

In scientific research, one of the keys to the method of data analysis is covariance, which reflects the joint variability of two random variables. In statistics, covariance is defined as the mean value of the product of the deviations of two random variables from their respective means. The application of covariance is extended to two directions. One is correlation coefficient, which can be treated as standardized covariance, and the other is correlation function, which can regarded as generalized covariance. A number of measures have been derived from correlation coefficient, and one of them is spatial autocorrelation coefficient. The typical spatial autocorrelation coefficient is Moran's index [1]. Based on spatial autocorrelation coefficients, spatial autocorrelation analysis methods has been developed. Spatial autocorrelation analysis originated from statistics and biometrics [1–3]. After the vigorous development of geographers [4–8], spatial autocorrelation analysis is now widely used in geography [9–12], biology [13, 14], ecology [15–18], economics [19–22], and many other fields [23–25]. On the other hand, correlation function analysis is associated with spectral analysis. Spectral analysis includes the methods of power spectrum for time series and wave spectrum for spatial series [26–29]. Today, correlation function is linked to multifractal analysis because the global fractal dimension is based on Renyi entropy and generalized correlation function [26, 30–36]. In theory, the spatial analyses based on correlation coefficients and those based on correlation functions should reach the same goal by different routes, and thus can be integrated into a logical framework.

However, how to establish the relationships between spatial autocorrelation coefficients and fractal dimensions is still not clear enough. Scientific analysis depends on mathematical modeling [37]. There are two well-known obstacles in mathematical modeling: one is spatial dimension [38], the other is time lag [27]. Unfortunately, both of these difficulties are encountered in spatial analysis of complex geographical systems. Geo-spatial data analyses rely heavily on spatial correlation, including autocorrelation and cross-correlation. The precondition of using traditional statistical methods to analyze spatial data is that there is no correlation between spatial sampling points. Otherwise, the probability structure of spatial samples is not determinate due to uncertain mean values, and thus the conventional statistical methods such as regression analysis and principal component analysis will be not credible. In this case, spatial autocorrelation modeling is always employed to make data analysis [4, 5, 7–9]. The common spatial autocorrelation measures include Moran's index [2], Geary's coefficient [3], and Getis-Ord's index [6]. Recently, a new spatial statistic called *B*-statistic, which is based on both geographical and covariate-proximity, was proposed for measuring spatio-environmental autocorrelation [24]. Unfortunately, due to spatial complexity, the values of spatial autocorrelation indexes are often depend on measurement scales. This suggests spatial scaling and fractal patterns. Maybe we can associate spatial autocorrelation coefficient with fractal parameters and find new approaches to making spatial analysis for complex systems such as cities. This paper is devoted to deriving the inherent association of spatial autocorrelation coefficient with spatial correlation dimension. In Section 2, the concepts and models of spatial correlation functions and spatial correlation dimension are clarified, and the then spatial correlation dimension is derived from spatial autocorrelation functions based on Moran's index. In

Section 3, to verify the theoretical results, the derived models are applied to the Chinese cities. In Section 4, the related questions are discussed. Finally, the discussion is concluded by summarizing the main points of this work.

## 2 Theoretical models

### 2.1 Spatial correlation dimension

In the process of spatial analysis, we encounter a paradox. This paradox may suggests the uncertainty principle of spatial correlation. If there is no spatial autocorrelation among a group of spatial elements, the spatial autocorrelation coefficient is reliable and it equals zero in theory. On the contrary, if there is significant spatial autocorrelation, the values of spatial autocorrelation indicators such as Moran's index will be incredible [39]. The calculation of the spatial correlation coefficient depends on the mean or even the standard deviation [23]. The mean is based on the sum of observational values. Spatial autocorrelation implies that the whole is not equal to the sum of its parts, and therefore the mean and standard deviation are not affirmatory. As a result, the value of spatial autocorrelation coefficients will significantly deviate from the confidence values. One way to solve the above problem is to take into account scaling process of spatial distributions. If so, we should associate spatial autocorrelation with fractal dimension. Among various fractal parameters, the correlation dimension is most likely to be related to the spatial autocorrelation function. In this case, we can derive spatial correlation dimension for geographical systems.

Correlation dimension is one of basic parameters of multifractal systems, and it is actually defined on the basis of Renyi entropy and correlation function. Correlation functions can be divided into two types: correlation density function and correlation sum function. The former is based on density distribution function, and the latter is based on cumulative distribution function [26]). In urban science, spatial correlation density function is also termed density-density correlation function, which can be expressed as follows

$$c(r) = \int_{-\infty}^{\infty} \rho(x)\rho(x+r)\mathrm{d}x, \tag{1}$$

where $c(r)$ refers to the density correlation, $\rho(x)$ denotes city density, $x$ is the location of a certain city (defined by the radius vector), and $r$ is the distance to $\boldsymbol{x}$ and it represents spatial displacement parameter. Spatial displacement in spatial analysis is equivalent to time lag in time series analysis. Both time lag and spatial displacement suggest response delay. In terms of Eq (1), if there is a city at $\boldsymbol{x}$, the probability to find another city at distance $r$ from $x$ is $c(r)$. The correlation function based on integral is useful in theoretical deduction. In application, the continuous form should be replaced by discrete form, which can be expressed as

$$c(r) = \frac{1}{S} \sum_x \rho(x)\rho(x+r), \tag{2}$$

where $S$ denotes the area of a geographical unit occupied by a system of cities. The other symbols are the same as those in Eq (1). If we can find the relationship between the correlation function $c(r)$ and the spatial displacement $r$, we can make a spatial analysis of cities (Fig 1). Eq (1) is the discrete expression of density-density correlation function. Through integral, it can be transformed into a correlation sum function as below [40, 41]:

$$C(r) = \frac{1}{S} \sum_x A(x)A(x+r), \tag{3}$$

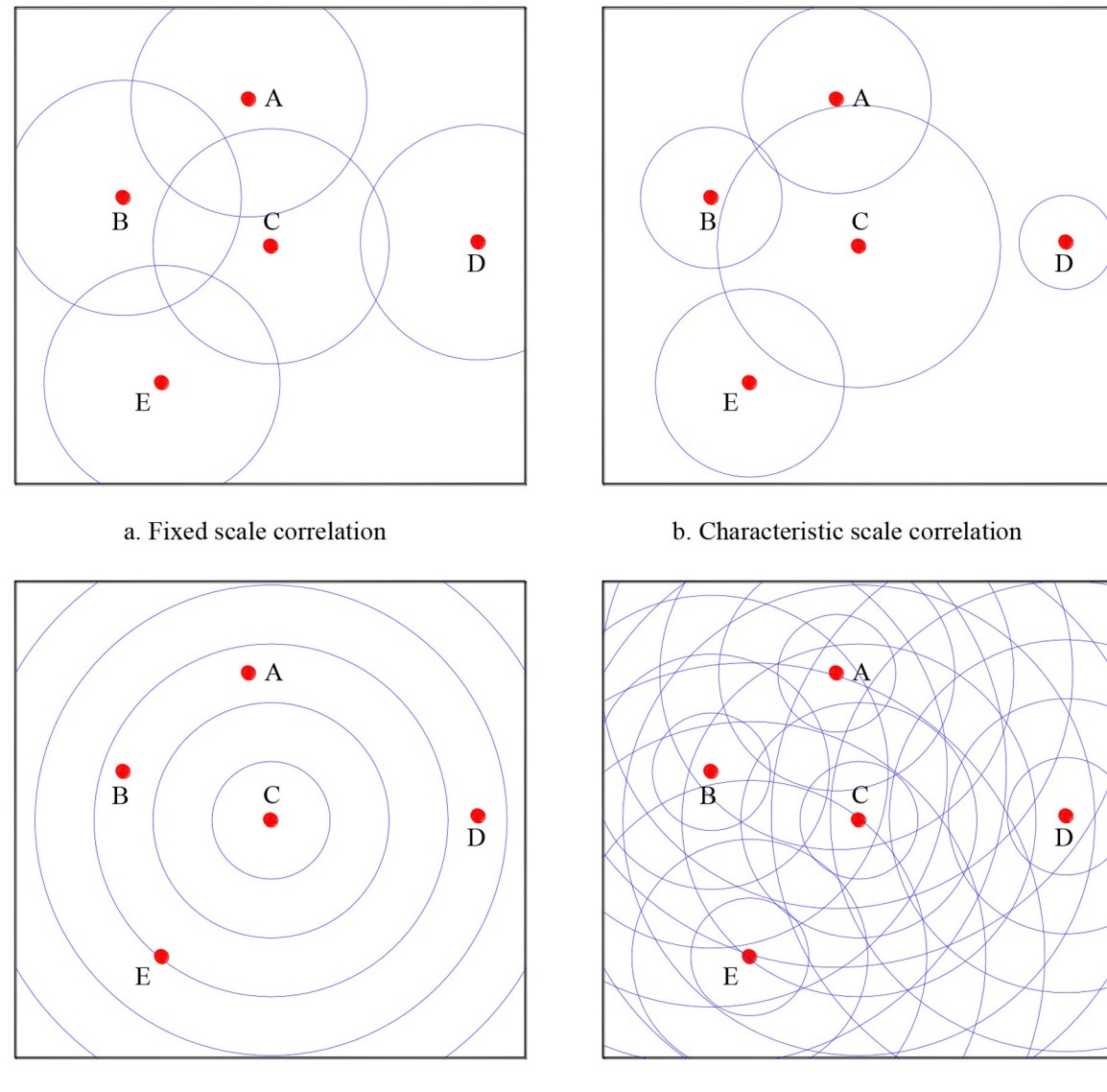

a. Fixed scale correlation

b. Characteristic scale correlation

c. Local scaling correlation

d. Global scaling correlation

**Fig 1. A sketch map of spatial correlation which fall in four types. Note:** The spatial correlation based on fixed scale can be used to calculate Moran's index, the one point correlation based on local scaling can be used to compute radial fractal dimension, and the point-point correlation based on global scaling can be used to calculate spatial correlation dimension and define spatial autocorrelation function.

where $C(r)$ is called *correlation integral* or *correlation sum* [42], $A(x)$ denotes urban mass. The density correlation is a decreasing function, while the mass correlation is an increasing function. Correlation density functions are susceptible to random perturbations. In contrast, cumulative function has strong anti-noise ability, and thus can better reflect the spatial regularity.

In practice, if we use the categorical (nominal) variable to substitute the metric variable, the correlation sum function can be further simplified. Based on spatial nominal variable, Eq (3)

can be rewritten as

$$C(r) = \frac{N(r)}{N^2} = \frac{1}{N^2} \sum_{i=1}^{N} \sum_{j=1}^{N} H(r - d_{ij}), \tag{4}$$

which $r$ refers to the yardstick indicative of distance threshold, $N$ denotes city number, $N(r)$ is the number of the cities have correlation, $d_{ij}$ is the distance between city $i$ and city $j$ ($i$, $j$ = 1,2,3,. . .,$N$), and $H()$ is the Heaviside function. The property of Heaviside function is as below

$$H(r - d_{ij}) = \begin{cases} 1, & \text{when } d_{ij} \leq r; \\ 0, & \text{when } d_{ij} > r. \end{cases} \tag{5}$$

This implies that $r$ forms a distance threshold by the Heaviside function. If the relationship between correlation sum and the distance threshold follow a power law such as

$$C(r) = C_1 r^{D_c}, \tag{6}$$

we will have a scale-free correlation, and $D_c$ is the correlation dimension coming between 0 and 2. In Eq (6), $C_1$ refers to the proportionality coefficient. In empirical analyses, the correlation sum $C(r)$ can be replaced by correlation number $N(r)$ to determine fractal dimension. Obviously, the correlation number is

$$N(r) = \sum_{i=1}^{N} \sum_{j=1}^{N} H(r - d_{ij}). \tag{7}$$

Then Eq (6) should be substituted with the following relation

$$N(r) = N^2 C(r) = N_1 r^{D_c}, \tag{8}$$

where $N_1 = C_1 N^2$ denotes the proportionality coefficient. Replacing the correlation function $C(r)$ with the correlation number $N(r)$ has no influences on the value of the spatial correlation dimension, $D_c$. In this case, Eq (8) is actually equivalent to Eq (6) in geographical spatial analysis.

## 2.2 Relation between correlation dimension and autocorrelation function

Generalizing spatial autocorrelation coefficients yields corresponding spatial autocorrelation functions. The spatial autocorrelation function can be regarded as a set of spatial autocorrelation coefficients [43]. Spatial autocorrelation coefficients are determined by size measures and spatial proximity measures. A spatial proximity matrix, $\mathbf{V}$, which is a spatial distance matrix or a spatial relation matrix, can be converted into a contiguity matrix as follows: $\mathbf{V} = [v_{ij}]$. The spatial contiguity can be defined by a relative step function as below

$$v_{ij}(r) = \begin{cases} 1, 0 < d_{ij} \leq r \\ 0, d_{ij} > r \end{cases}, \tag{9}$$

where $d_{ij}$ refers to the distance between locations $i$ and $j$, $r$ denotes a variable distance threshold. For the diagonal elements ($i = j$), if $d_{ii} = 0$ suggests $v_{ii}(r) = 0$, then we will have $\mathbf{M}(r) = [v_{ij}(r)]_{N*N}$. This is one basis for conventional spatial autocorrelation analysis. On the other, for $i = j$, if $d_{ii} = 0$ suggests $v_{ii}(r) = 1$, then we will have $\mathbf{M}^*(r) = [v_{ij}(r)]_{N*N}$. This will be used to make

generalized spatial autocorrelation analysis. Obviously, the difference between $\mathbf{M}^*(r)$ and $\mathbf{M}(r)$ is a unit matrix $\mathbf{E}$, that is $\mathbf{M}^*(r) - \mathbf{M}(r) = \mathbf{E}$.

The sum of the elements in the contiguity matrix is as follows

$$T(r) = \sum_{i=1}^{n} \sum_{j=1}^{n} v_{ij}(r) = \begin{cases} M_0(r), & v_{ii} = 0 \\ M_0^*(r), & v_{ii} = 1 \end{cases}. \tag{10}$$

Define an ones vector $\mathbf{o} = [1, 1, \ldots, 1]^T$, we have $M_0(r) = \mathbf{o}^T \mathbf{M}(r)\mathbf{o}$, $M_0^*(r) = \mathbf{o}^T \mathbf{M}^*(r)\mathbf{o}$. Apparently, $N = n = \mathbf{o}^T \mathbf{E}\mathbf{o}$. Thus the number of non-zero elements in the matrix $\mathbf{M}(r)$ is

$$N(r) = M_0^*(r) = M_0(r) + N = \mathbf{o}^T \mathbf{M}^*(r)\mathbf{o}. \tag{11}$$

According to Eq (7), $N(r)$ is just the correlation number of cities. In order to unitize the spatial contiguity matrix, define

$$\frac{v_{ij}(r)}{T(r)} = \frac{v_{ij}(r)}{\sum_{i=1}^{n} \sum_{j=1}^{n} v_{ij}(r)} = \begin{cases} w_{ij}(r), & v_{ii}(r) = 0 \\ w_{ij}^*(r), & v_{ii}(r) = 1 \end{cases}. \tag{12}$$

Thus we have $\mathbf{W}(r) = [w_{ij}(r)]_{n*n}$, $\mathbf{W}^*(r) = [w_{ij}^*(r)]_{n*n}$. With the preparation of the above definitions and symbolic system, we can define the spatial autocorrelation function. Based on standardized size vector $\mathbf{z}$ and global unitized spatial weight matrix $\mathbf{W}$, Moran's index of spatial autocorrelation can be expressed as [23].

$$I = \mathbf{z}^T \mathbf{W}\mathbf{z}. \tag{13}$$

Replacing the determined unitized spatial weight matrix $\mathbf{W}$ by the variable unitized spatial weight matrix $\mathbf{W}(r)$ yields

$$I(r) = \mathbf{z}^T \mathbf{W}(r)\mathbf{z}, \tag{14}$$

which is a spatial autocorrelation function of displacement based on Moran's index [43].

The conventional spatial autocorrelation coefficient, Moran's $I$, is obtained by analogy with the temporal autocorrelation function in the theory of time series analysis. For time series analysis, if time lag is zero ($\tau = 0$), the autocorrelation coefficient reflects the self-correlation of a variable at time $t$ to the variable at time $t$. In this case, the autocorrelation coefficient must be equal to 1, a known number, and thus yields no any useful information. As a result, the zero time lag is not taken into account in time series analysis. The diagonal elements of the space contiguity matrix correspond to the zero lag of the time series. Accordingly, the values of the diagonal elements of the spatial contiguity matrix is always set as 0. As a matter of fact, the diagonals represent the self-correlation of spatial elements in a geographical system, e.g., city A with city A, city B with city B. This kind of correlation cannot be ignored in many cases. If we consider the zero-lag self-correlation of geographical elements, Moran's index can be generalized to the following form

$$I^*(r) = \mathbf{z}^T \mathbf{W}^*(r)\mathbf{z}. \tag{15}$$

In the spatial weight matrix $\mathbf{W}^*(r)$, the values of the diagonal elements are 1. In short, spatial autocorrelation differs from temporal autocorrelation, and the diagonal elements of spatial contiguity matrix can be taken into consideration in some cases.

If a geographical process of spatial autocorrelation has characteristic scales, we will have certain values of Moran's index. In this instance, the spatial correlation function is not

necessary. On the contrary, if a geographical correlation process bears no characteristic scale, the spatial autocorrelation function suggests scaling process in the geographical pattern. Scaling invariance is one of necessary conditions for fractal structure. Thus, maybe we can find the fractal properties in a spatial autocorrelation process. Based on the concepts of spatial correlation functions and spatial autocorrelation functions, the relations between Moran's index and fractal dimension can be derived. After a series of mathematical reasoning and transformation (S1 File), the relationship between Moran's index and spatial correlation dimension can be derived as follows

$$I^*(r) - \frac{I(r)}{1 + N/M_0(r)} = \frac{N}{N(r)} = \frac{N}{N_1} r^{-D_c}, \tag{16}$$

which gives the mathematical relationships between the spatial autocorrelation function, $I(r)$, the generalized autocorrelation function, $I^*(r)$, and the spatial correlation dimension, $D_c$. Considering Eq (4), $C(r) = N(r)/N^2$, we have a spatial correlation equation such as

$$\frac{1}{C(r)} = NI^*(r) - \frac{N}{1 + N/M_0(r)} I(r) = \frac{N^2}{N_1} r^{-D_c}. \tag{17}$$

With the increase of $r$, $N/M_0(r)$ approaches 0. Thus, for large spatial datasets, we have an approximate expression as below:

$$\Delta I(r) = I^*(r) - I(r) \approx \frac{N}{N(r)} = \frac{1}{NC(r)} = \frac{N}{N_1} r^{-D_c} = \frac{1}{NC_1} r^{-D_c}, \tag{18}$$

where $\Delta I(r)$ denotes the difference between $I^*(r)$ and $I(r)$, and the parameter $C_1 = N_1/N^2$.

Up to now, we have derived the exact and approximate relationships between spatial correlation dimension and spatial autocorrelation function. The spatial correlation function comprises a series of spatial autocorrelation coefficients based on Moran's index. Using observational data, we can testify the main relations derived from the theoretical principle of spatial correlation processes.

## 3 Empirical analysis

### 3.1 Datasets and methods

The goal of this study is to establish a mathematical model that connects spatial autocorrelation and spatial scaling laws. A as Neumann [44] once said: "The sciences do not try to explain, they hardly even try to interpret, they mainly make models." In this sense, this work relies heavily on mathematical derivation and modeling rather than data analysis. After all, as Karlin [45] pointed out: "The purpose of models is not to fit the data, but to sharpen the questions." Despite this, it is significant to verify a modeling result by observational data. I am very much in favor of his viewpoint of Louf and Barthelemy [46], who said: "The success of natural sciences lies in their great emphasis on the role of quantifiable data and their interplay with models. Data and models are both necessary for the progress of our understanding: data generate stylized facts and put constraints on models. Models on the other hand are essential to comprehend the processes at play and how the system works. If either is missing, our understanding and explanation of a phenomenon are questionable. This issue is very general, and affects all scientific domains, including the study of cities." Therefore, the network of Chinese cities can be employed to verify the models derived in last section. For comparability and simplifying the analytical processes, only municipalities directly under the Central Government of China and provincial capitals are taken into account in this case. There are 31 provinces,

municipalities, and autonomous regions in Chinese mainland. So, this network includes 31 large cities. Basic data include urban population and railway mileage. Urban population represents city size measure, while the spatial contiguity matrix is generated by railway distances. Population data came from the fifth (2000) and sixth (2010) censuses, and railway mileage came from China's traffic mileage map. However, two cities, Lhasa and Haikou, were not connected to the network by railway for a long time. Therefore, only 29 cities compose the spatial sample ($N = 29$) (see files S1 and S2 Datasets).

The analytical procedure can be outlined according to the theoretical derivation process. The computational steps are as follows.

**Step 1: define the yardsticks of spatial correlation.**   The yardstick is a variable of distance threshold, which is designed in light of the railway mileage matrix. Its function bears analogy with time lag parameter in time series analysis. If the zero elements on the diagonal are overlooked, the minimum traffic mileage is 137 kilometer and the maximum traffic mileage is 5062 kilometer. So the yardstick length can be taken as $r = 150, 250, 350, . . ., 5150$.

**Step 2: calculate spatial correlation function.**   Using Heaviside function, Eq (5), we can obtain spatial correlation number $N(r)$, and spatial correlation function, $C(r)$. Based on scaling range, the correlation dimension can be evaluated by the power law relation between the yardstick length $r$ and spatial correlation number $N(r)$ or spatial correlation function $C(r)$.

**Step 3: compute spatial autocorrelation measurements based on variable yardstick.** The spatial autocorrelation measures include Moran's index, Geary's coefficient, and Getis-Ord's index. This work is mainly based on Moran's index, supplemented by Geary coefficient and Getis-Ord's index.

**Step 4: verify the relationship between spatial autocorrelation measures and fractal dimension.**   Using Eq (16) and (18), we can confirm the relationships between Moran's index and spatial correlation dimension. In theory, this positive study can be generalized to the relationships between fractal dimension and Geary's coefficient and Getis-Ord's index.

Analytical process and results depend heavily on the definition and structure of spatial weight matrix. Where structure is concerned, two aspects of factors significantly influence analytical ways. One is diagonal elements, and the other is sum of spatial contiguity matrix. For fractal analysis, the diagonal elements should be taken into account, while for conventional spatial autocorrelation analysis, the diagonal elements should be removed. For generalized spatial autocorrelation analysis, the diagonal elements can be taken into consideration, while for special fractal analysis, the diagonal element can be deleted. On the other hand, for practical spatial autocorrelation function, the sum of spatial contiguity matrix should be fixed to the original sum value. However, for theoretical spatial autocorrelation function, the sum varies with the yardstick length. Different sums of spatial contiguity matrix plus different diagonal elements lead to four approaches to spatial correlation dimension and autocorrelation analyses (Table 1).

## 3.2 Computed results and analysis

Using the above-mentioned data and methods shown above, we can testify the models proposed in this work. In fractal analysis, scaling relationships take on two forms: one is global scaling invariance, and the other is local scaling invariance. The global scaling relations imply that all data points follow power law and form a straight line on the double logarithmic plot. In contrast, the local scaling relations indicate that only part data points follow power law and form a local straight line segment on the log-log plot. In theory, all the scaling relations are global power law relations, but empirically, almost all scaling relationships are local power law relations. In many cases, if the linear scale for measurement is too large or too small, the power

**Table 1. Four types of calculation approaches to spatial autocorrelation measurements.**

|  | Variable sum of distance matrix [V] | Fixed sum of distance matrix [F] |
|---|---|---|
| **All elements (including diagonal elements) [D]** | [D+V] Generalized Moran's function, $I^*(r)$ | [D+F] Generalized Moran's function, $I_f^*(r)$ |
| **Partial elements (excluding diagonal elements) [N]** | [N+V] Conventional Moran's function, $I(r)$ | [N+F] Conventional Moran's function, $I_f(r)$ |
| **Application direction** | Theoretical study and fractal analysis | Practical study and spatial autocorrelation analysis |

law relations break [47]). The local straight line segment represents the scaling range for fractal analysis. Partial calculation results are tabulated as below (Table 2). If the yardstick length is less than 300 milometers or greater than 2700 milometers, the power law relations break. The scaling range varies from 350 milometers to 2650 milometers. The relation between yardstick length $r$ and the correlation number $N(r)$ follows the power law, and the mathematical model based on theoretical derivation result is as follows

$$\hat{N}(r) = 0.0153r^{1.3623}. \tag{19}$$

The parameter values of the model are estimated by double logarithmic linear regression based on the least squares method. Diagonal elements of the correlation matrix were taken into account in the calculation process. The goodness of fit is about $R^2 = 0.9965$, and the spatial correlation dimension is about $D_2 = 1.3623 \pm 0.0358 < 1.5$ (Fig 2(A)). The symbol "^" denotes that the result is estimated value. The critical value of spatial correlation dimension is 1.5. The above result suggests weak spatial autocorrelation of Chinese cities. If the diagonal elements are removed so that the correlation matrix corresponds to the spatial weight matrix of spatial autocorrelation, the model of spatial scaling relation is as below:

$$\hat{N}^*(r) = 0.0012r^{1.6869}. \tag{20}$$

The goodness of fit is about $R^2 = 0.9936$, and the spatial correlation dimension is about $D_2^*$ $= 1.6869 \pm 0.0600$ (Fig 2(B)). This result is close to the empirical expected value of fractal dimension of cities, that is, 1.7 [48]. Eq (19) is based on Eq (6) and can be derived from the principle of spatial correlation based on scaling law, while Eq (20) represents an approximate empirical scaling relation for spatial correlation. If and only if the number of elements is infinite, i.e., $N \rightarrow \infty$, there is no difference between the two equations. However, there are still many scholars who prefer to use the empirical relationship rather than the theoretical model to estimate the correlation dimension [49–51].

The spatial correlation dimension has been theoretically associated with spatial autocorrelation functions based on conventional Moran's indexes and generalized Moran's indexes. This relation can be verified by Eq (16) or Eq (17). For the dataset in 2000, the mathematical model is as below:

$$\frac{1}{N\hat{C}(r)} = \hat{I}^*(r) - \frac{1}{1 + N/M_0(r)}\hat{I}(r) = 1893.8457r^{-1.3623}. \tag{21}$$

The coefficient of determination is about $R^2 = 0.9965$, and the spatial correlation dimension is around $D_c = D_2 = 1.3623$. Eq (21) can be transformed into the following correlation model

$$\hat{C}(r) = \frac{r^{1.3623}}{1893.8457N} = \frac{0.0153}{N^2}r^{1.3623}. \tag{22}$$

which is equivalent to Eq (19). Where spatial correlation function is concerned, this is the dimension estimation value based on an exact relation. Then, the 2010 urban census data is

**Table 2. Datasets for spatial correlation dimension and spatial autocorrelation analysis (Partial results).**

| Scale | Number | | 2000 (Fifth census data) | | | | 2010 (Sixth census data) | | | |
|---|---|---|---|---|---|---|---|---|---|---|
| $r$ | $N(r)$ | $N^*(r)$ | Moran $I^*$ | Moran $I$ | $\Delta I$ | $1/(NC(r))$ | Moran $I^*$ | Moran $I$ | $\Delta I$ | $1/(NC(r))$ |
| **150** | 31 | 2 | 1.0411 | 1.6363 | -0.5953 | 0.9355 | 1.1172 | 2.8164 | -1.6992 | 0.9355 |
| **250** | 39 | 10 | 0.8015 | 0.2257 | 0.5758 | 0.7436 | 0.9139 | 0.6643 | 0.2496 | 0.7436 |
| **350** | 49 | 20 | 0.5907 | -0.0028 | 0.5935 | 0.5918 | 0.6931 | 0.2481 | 0.4450 | 0.5918 |
| **450** | 63 | 34 | 0.4130 | -0.0877 | 0.5007 | 0.4603 | 0.5008 | 0.0749 | 0.4258 | 0.4603 |
| **550** | 85 | 56 | 0.2876 | -0.0813 | 0.3689 | 0.3412 | 0.3303 | -0.0164 | 0.3468 | 0.3412 |
| **650** | 103 | 74 | 0.2158 | -0.0915 | 0.3073 | 0.2816 | 0.2670 | -0.0203 | 0.2892 | 0.2816 |
| **750** | 127 | 98 | 0.1681 | -0.0780 | 0.2462 | 0.2283 | 0.1948 | -0.0435 | 0.2383 | 0.2283 |
| **850** | 139 | 110 | 0.1065 | -0.1291 | 0.2356 | 0.2086 | 0.1215 | -0.1101 | 0.2316 | 0.2086 |
| **950** | 155 | 126 | 0.1080 | -0.0972 | 0.2053 | 0.1871 | 0.1250 | -0.0764 | 0.2014 | 0.1871 |
| **1050** | 187 | 158 | 0.0489 | -0.1257 | 0.1746 | 0.1551 | 0.0543 | -0.1193 | 0.1736 | 0.1551 |
| **1150** | 209 | 180 | 0.0478 | -0.1056 | 0.1534 | 0.1388 | 0.0471 | -0.1064 | 0.1535 | 0.1388 |
| **1250** | 255 | 226 | 0.0668 | -0.0529 | 0.1197 | 0.1137 | 0.0471 | -0.0752 | 0.1223 | 0.1137 |
| **1350** | 295 | 266 | 0.0357 | -0.0695 | 0.1051 | 0.0983 | 0.0314 | -0.0742 | 0.1056 | 0.0983 |
| **1450** | 329 | 300 | 0.0312 | -0.0624 | 0.0936 | 0.0881 | 0.0199 | -0.0748 | 0.0947 | 0.0881 |
| **1550** | 353 | 324 | 0.0717 | -0.0113 | 0.0831 | 0.0822 | 0.0643 | -0.0194 | 0.0837 | 0.0822 |
| **1650** | 381 | 352 | 0.0491 | -0.0293 | 0.0783 | 0.0761 | 0.0471 | -0.0314 | 0.0785 | 0.0761 |
| **1750** | 397 | 368 | 0.0372 | -0.0387 | 0.0759 | 0.0730 | 0.0359 | -0.0400 | 0.0760 | 0.0730 |
| **1850** | 437 | 408 | 0.0491 | -0.0185 | 0.0676 | 0.0664 | 0.0431 | -0.0250 | 0.0684 | 0.0664 |
| **1950** | 471 | 442 | 0.0348 | -0.0285 | 0.0633 | 0.0616 | 0.0331 | -0.0303 | 0.0634 | 0.0616 |
| **2050** | 501 | 472 | 0.0408 | -0.0182 | 0.0589 | 0.0579 | 0.0376 | -0.0215 | 0.0591 | 0.0579 |
| **2150** | 547 | 518 | 0.0179 | -0.0371 | 0.0550 | 0.0530 | 0.0151 | -0.0401 | 0.0551 | 0.0530 |
| **2250** | 575 | 546 | 0.0043 | -0.0486 | 0.0529 | 0.0504 | 0.0005 | -0.0526 | 0.0531 | 0.0504 |
| **2350** | 611 | 582 | 0.0217 | -0.0271 | 0.0487 | 0.0475 | 0.0176 | -0.0313 | 0.0490 | 0.0475 |
| **2450** | 633 | 604 | 0.0045 | -0.0433 | 0.0478 | 0.0458 | 0.0042 | -0.0436 | 0.0478 | 0.0458 |
| **2550** | 667 | 638 | 0.0175 | -0.0271 | 0.0447 | 0.0435 | 0.0171 | -0.0276 | 0.0447 | 0.0435 |
| **2650** | 685 | 656 | 0.0095 | -0.0343 | 0.0438 | 0.0423 | 0.0093 | -0.0345 | 0.0438 | 0.0423 |
| **2750** | 699 | 670 | 0.0047 | -0.0384 | 0.0431 | 0.0415 | 0.0030 | -0.0401 | 0.0432 | 0.0415 |
| **2850** | 709 | 680 | 0.0022 | -0.0403 | 0.0426 | 0.0409 | 0.0007 | -0.0420 | 0.0426 | 0.0409 |
| **2950** | 717 | 688 | 0.0026 | -0.0394 | 0.0420 | 0.0404 | 0.0019 | -0.0402 | 0.0421 | 0.0404 |
| **3050** | 729 | 700 | -0.0053 | -0.0470 | 0.0416 | 0.0398 | -0.0053 | -0.0470 | 0.0416 | 0.0398 |

**Note:** (1) Only partial results are displayed in this table. More results are attached in the Supporting Information files. (2) Moran's index comes between -1 and 1, otherwise the results are outliers. Corresponding to the yardstick length $r$ = 150, several Moran's index values are abnormal and can be treated as outliers. (3) The main model parameters appearing in this section were estimated by using the data in this table. The relationship between $r$ and $N(r)$ gives equation (19), the relationship between $r$ and $N^*(r)$ gives Eq (20), the relationship between $r$ and $1/(NC(r))$ gives Eq (21), which suggests Eq (22). $\Delta I$ is the difference between $I(r)$ and $I^*(r)$. The relationships between $r$ and $\Delta I$ give Eq (23) and (24).

used to replace the 2000 urban census data, and the calculation results remain unchanged (Fig 3). The reason is that the spatial weight matrix has not changed. This suggests that the spatial scaling exponent of Eq (16) or Eq (17) depend on spatial contiguity matrix rather than urban population sizes. Spatial correlation dimension is only determined by spatial patterns. In this sense, for spatial complex systems, element size is a fast variable, while spatial contiguity is a slow variable. For self-organizing cities, slow variables dominate fast variables [52].

If the spatial correlation number is significantly greater than the city number, the exact relation between Moran's function and yardstick length can be replaced by an approximate relation. Through Eq (18), we can verify this approximate scaling relation (Fig 4). For the dataset

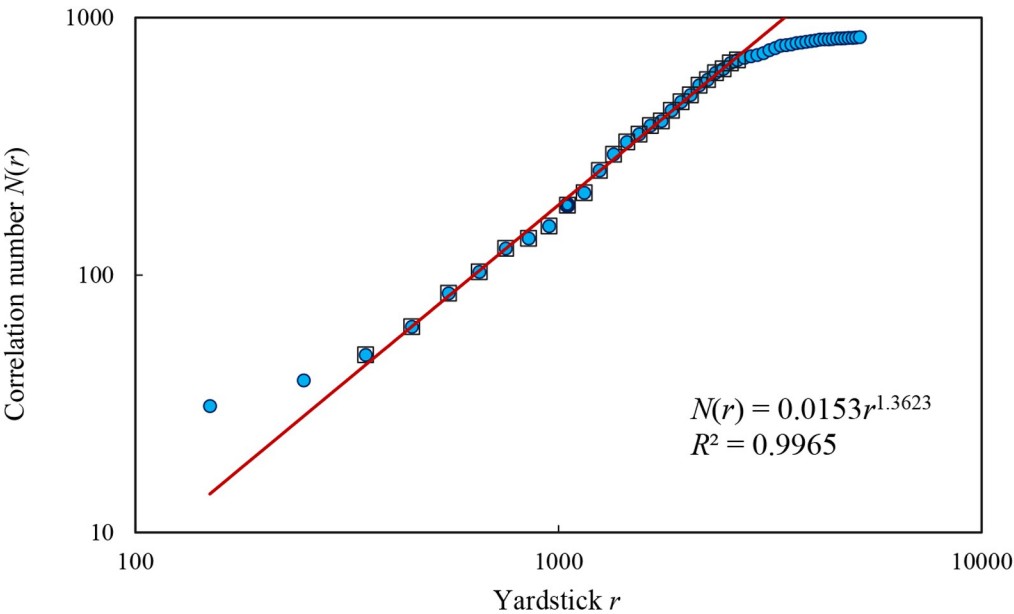

a. Theoretical relation: Include diagonal elements

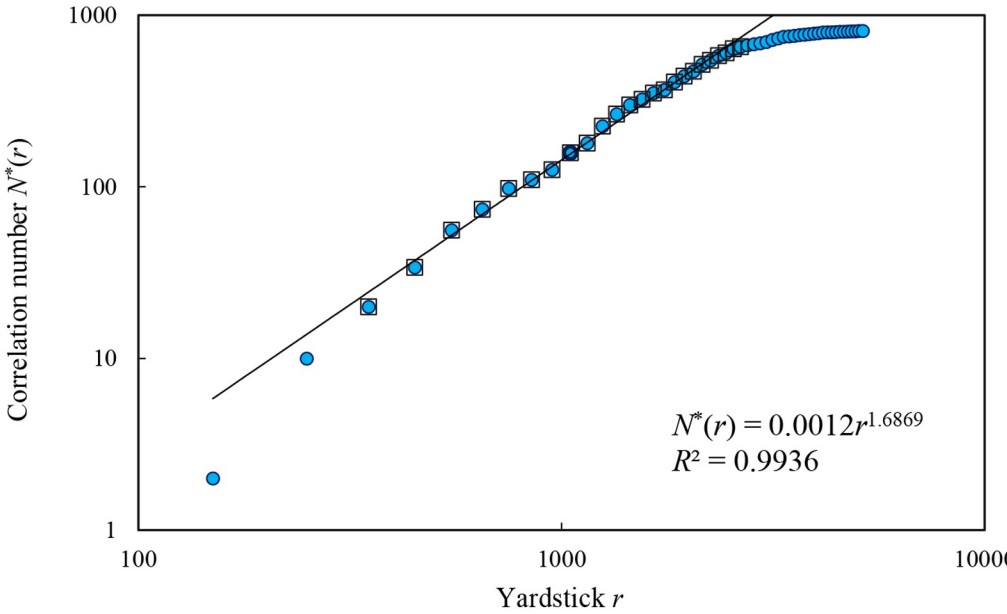

b. Empirical relation: Exclude diagonal elements

**Fig 2. The scaling relation for spatial correlation dimension of Chinese capital cities based on railway distance.**
**Note:** The solid dots represent the total number of spatial correlations, and the hollow blocks represent the points within the scaling range. The latter is a subset of the former.

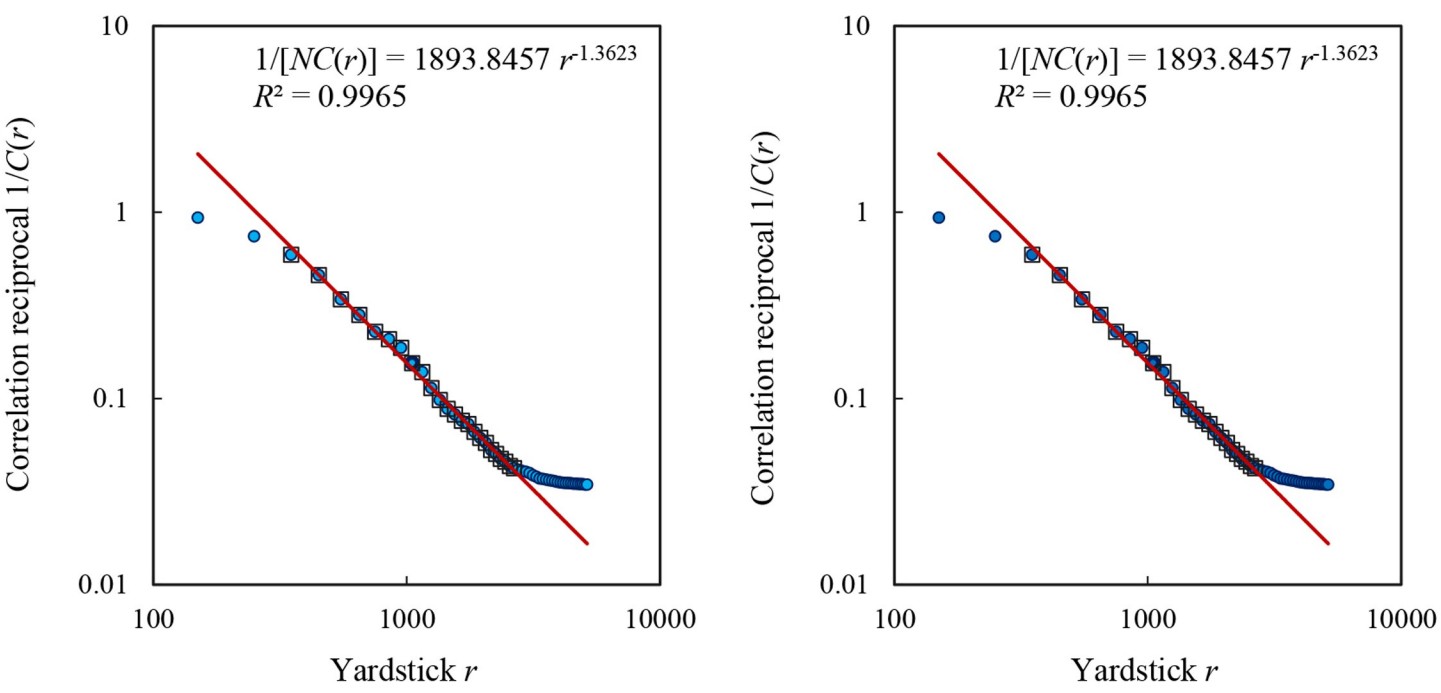

**Fig 3. The scaling relations for the reciprocal of spatial correlation function based on Moran's function.** a. 2000. b. 2010. **Note:** The solid dots represent the total number of spatial autocorrelation functions, and the hollow blocks represent the points within the scaling range. The scaling range corresponds to that in Fig 2.

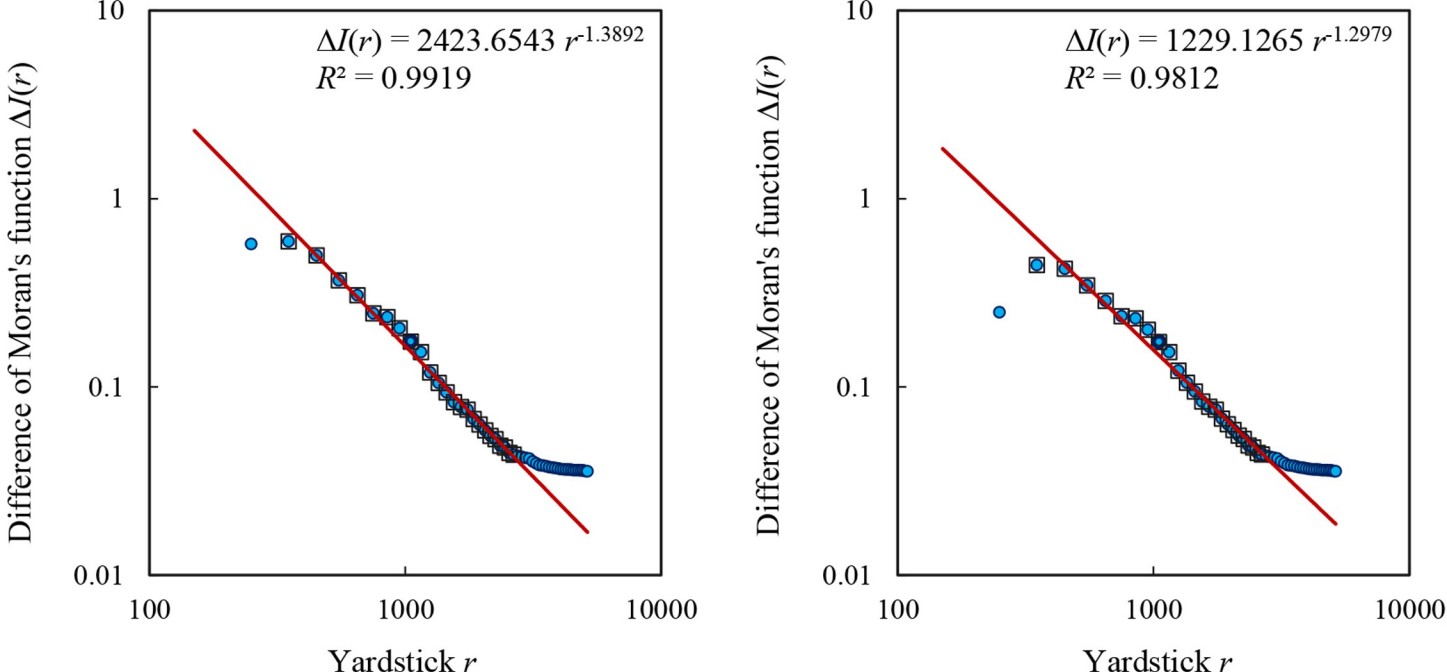

**Fig 4. The scaling relations for the difference between two types of Moran's functions.** a. 2000. b. 2010. **Note:** The solid dots represent the total number of difference of Moran's functions, and the hollow blocks represent the points within the scaling range. The scaling range is consistent with those in Figs 1 and 2.

from 2000, the model based on the least square calculation is as follows

$$\Delta\hat{I}(r) = 2423.6543r^{-1.3892}. \tag{23}$$

The goodness of fit is about $R^2 = 0.9919$, and the spatial correlation dimension is estimated as about $D_c = 1.3892$. For the data from 2010, the model is as below:

$$\Delta\hat{I}(r) = 1229.1265r^{-1.2979}. \tag{24}$$

The goodness of fit is about $R^2 = 0.9812$, and the spatial correlation dimension is about $D_c = 1.2979$. The goodness of fit decrease, and the fractal dimension estimation results departed from the expected value. In this case, both urban population sizes and spatial contiguity matrix influence the parameter estimation values.

This study is devoted to exploring the theoretical relationships between spatial autocorrelation and spatial dimension. The aim is at revealing the scaling law in spatial autocorrelation processes. The positive analysis of spatial autocorrelation and fractal dimension of urban systems is not the main task of this work. Based on the above calculation results, the inferences can be made as follows. First, spatial correlation dimension depends on spatial contiguity matrix. It is independent of size measures. Even if the city sizes changes, but the spatial distances between cities does not change, then the spatial correlation dimension remains unchanged. In this case, the relationships between Moran's function and spatial correlation dimension do not change. Second, the difference between common Moran's function and generalized Moran's function relies on both spatial contiguity matrix and size measures. If the number of cities in a region is large enough, the difference between the two types of Moran functions can be used to take place of the reciprocal of the correlation function. This relationships between the function difference values and yardstick lengths follow power law and give spatial correlation dimension approximately. In this instance, the spatial correlation dimension value is sensitive to the city sizes. A conclusion can be drawn that theoretical spatial correlation dimension depends on the patterns of spatial distribution rather size distribution. However, if we estimate the correlation dimension using the approximate formula, the result can be impacted by the size measure.

## 4 Discussion

Scientific research begins with description of a phenomenon and ends with understanding its essence. Accurate and quantitative scientific descriptions rely on mathematical methods. Mathematical methods in scientific research fall into two categories: one is based on characteristic scale, and the other, based on scaling invariance (Table 3). The key of conventional mathematical modeling and quantitative analysis are to catch characteristic scales [27, 53]. However, for complex systems such as cities, we cannot find characteristic scale in many cases [30]. In this case, an advisable method is to replace characteristic scale with scaling exponent. Fractal geometry provides a powerful tool for scaling analysis. Moran's index is an eigenvalue of generalized spatial correlation matrix and represents a characteristic scale of spatial autocorrelation [43]. Unfortunately, for complex spatial process, Moran's index depends on spatial proximity and cannot serve as a real characteristic scale. It is necessary to extend spatial autocorrelation coefficient to spatial autocorrelation function. The creation of the spatial autocorrelation function analysis method was inspired by the spatial correlation graph [43], which is referred to as *correlogram* in the literature [9, 20, 24, 54–57]. The correlogram can be used to detect the characteristic scale of spatial autocorrelation under some conditions [57]. A discovery is that, for the complex systems without characteristic scale, spatial autocorrelation

**Table 3. Comparison of two types of mathematical methods in geospatial analysis.**

| Type | General mathematical method | Special mathematical method |
|---|---|---|
| Basis | Characteristic scale | Scaling invariance |
| Mathematics | Euclidean Geometry, Calculus, Linear Algebra, Probability Theory and Statistics, Topology, and so on | Fractal geometry, wave-spectral analysis, wavelet analysis, allometric scaling theory, complex network theory, and so on |
| Key factor | Length, radius, average value, standard deviation, eigenvalue, etc | Scaling exponent, fractal dimension, rank-size exponent, etc. |
| Criterion | The measurement results do not depend on scale | The measurement results depend on scale |
| Application | Find characteristic scale | Determine scaling relation |
| Example | Moran's index represents an eigenvalue of spatial correlation matrix | Moran's index represents an eigenvalue of spatial correlation matrix |

**Note**: If length, area, average value, eigenvalue and so on do not depend on measurement scale, sample size, and so on, they represent characteristic length and can be used make models and conduct quantitative analysis. Otherwise, conventional mathematical methods will fail to a degree, and characteristic scale should be substituted with scaling relation. Moran's index is a kind of eigenvalue, and spatial correlation dimension belong to a scaling exponent.

function is related to scaling invariance within certain scale range [43]. The scaling invariance can be characterized by spatial correlation dimension. Spatial correlation dimension is a special scaling exponent, and it can be employed to complement the spatial autocorrelation function based on Moran's index for complex spatial processes and patterns.

Spatial correlation dimension is one of basic multifractal parameters. Multifractal scaling represents an important approach to describing a broad range of heterogeneous phenomena [58]. The classic theme of geography is *areal differentiation*, which can also be regarded as one of basic spatial effects caused by location [59]. Areal differentiation leads to spatial difference in geographical systems such as networks of cities. Thus, spatial correlation dimension provides an effective quantitative description of spatial heterogeneity in geographical research. In fact, multifractal dimension spectrum is definition based on both entropy and correlation function [31]. Entropy can be used to reflect spatial heterogeneity, while correlation function can be utilized to reflect spatial dependence. Spatial heterogeneity and spatial dependence are two central notions of geographical analysis [21, 59]. In this sense, spatial correlation dimension is helps to integrate spatial heterogeneity analysis and spatial dependence analysis into a logical framework. The above mathematical process suggests that, based on the relative step function of distance, spatial autocorrelation coefficients can be generalized to spatial autocorrelation functions. The typical spatial autocorrelation coefficient is Moran's index. The spatial autocorrelation function on the basis of Moran's index can be expressed as Eq (14). Taking into account the self-correlation of geographical elements, which appear on the diagonal of the spatial correlation matrix, the standard spatial autocorrelation function can be extended to the form of equation (15). Eqs (14) and (15) proved to be associated with the reciprocal of spatial correlation functions. The spatial correlation dimension $D_c$ can be derived from the standard spatial autocorrelation function $I(r)$ and the generalized spatial autocorrelation function, $I^*(r)$. Thus, the mathematical relationships between fractal dimension, autocorrelation coefficients, and spatial correlation dimension have been brought to light.

Moreover, the spatial correlation dimension can be linked to Geary's coefficient and Getis-Ord's index. The relationship between Moran's index and Geary's coefficient can be demonstrated as

$$C = \frac{n-1}{n}(\mathbf{o}^T\mathbf{W}\mathbf{z}^2 - \mathbf{z}^T\mathbf{W}\mathbf{z}) = \frac{n-1}{n}(\mathbf{e}^T\mathbf{W}\mathbf{z}^2 - I), \tag{25}$$

where $\mathbf{o} = [1\ 1\ \ldots\ 1]^T$, $\mathbf{z}^2 = [z_1^2\ z_2^2\ \ldots\ z_n^2]^T$. Introducing the spatial displacement parameter $r$

into Eq (25) yields the autocorrelation functions based on Geary's coefficient as follows

$$C_g(r) = \frac{n-1}{n}[\mathbf{o}^{\mathrm{T}}\mathbf{W}(r)\mathbf{z}^2 - I(r)], \tag{26}$$

where $C_g(r)$ denotes Geary's function, and the right subscript $g$ is used to differentiate Geary's function from spatial correlation function. Considering Eqs (8) and (11), and then rewriting Eq (16) yields

$$I(r) = \frac{M_0^*(r)}{M_0(r)}\left(I^*(r) - \frac{N}{N_1}r^{-D_c}\right) = \frac{1}{M_0(r)}(I^*(r)N_1 r^{D_c} - N). \tag{27}$$

Substituting Eq (27) into Eq (26) yields spatial correlation equation based on Geary's coefficient, that is

$$C_g(r) = \frac{n-1}{n}\left[\mathbf{o}^{\mathrm{T}}\mathbf{W}(r)\mathbf{z}^2 + \frac{N}{M_0(r)} - \frac{N_1 I^*(r)}{M_0(r)}r^{D_c}\right], \tag{28}$$

which gives the relationships between the spatial autocorrelation function based on Geary's coefficient and spatial correlation dimension $D_c$. If $n$ is large enough, then $(n\text{-}1)/n$ is close to 1 and $N/M_0(r)$ approaches 0, and Eq (28) can be replaced by an approximation relation.

Further, we can derive the relationship between Getis-Ord's index and spatial correlation dimension. Substituting the standardized size vector, $\mathbf{z}$, in Eq (13) with the global normalized size vector based on sum, $\mathbf{u}$, we can transform the formula of the spatial autocorrelation function based on Moran's index into that of Getis-Ord's index as follows

$$G(r) = \mathbf{u}^{\mathrm{T}}\mathbf{W}(r)\mathbf{u} = \frac{1}{M_0(r)}(\mathbf{u}^{\mathrm{T}}\mathbf{M}(r)\mathbf{u} - \mathbf{u}^{\mathrm{T}}\mathbf{u}). \tag{29}$$

Then, replacing $\mathbf{W}(r)$ with $\mathbf{W}^*(r)$, we can generalized standard spatial autocorrelation function to the following form

$$G^*(r) = \mathbf{u}^{\mathrm{T}}\mathbf{W}^*(r)\mathbf{u} = \frac{\mathbf{u}^{\mathrm{T}}\mathbf{u}}{N(r)} + \frac{M_0(r)G(r)}{N(r)} = \frac{\mathbf{u}^{\mathrm{T}}\mathbf{E}\mathbf{u} + \mathbf{u}^{\mathrm{T}}\mathbf{M}(r)\mathbf{u}}{\mathbf{o}^{\mathrm{T}}\mathbf{M}^*(r)\mathbf{o}}, \tag{30}$$

in which $\mathbf{u}^{\mathrm{T}}\mathbf{u}$ is a constant. Similar to the process of derivation of the relationships between Moran's index and spatial correlation dimension, a relation between Getis-Ord's index $G$ and fractal dimension $D_c$ can be derived as

$$G^*(r) - \frac{G(r)}{1 + N/M_0(r)} = \frac{\mathbf{u}^{\mathrm{T}}\mathbf{u}}{N(r)} = \frac{\mathbf{u}^{\mathrm{T}}\mathbf{u}}{N_1}r^{-D_c}, \tag{31}$$

where $N(r) = N^2 C(r)$. Eq (31) represents a spatial correlation equation based on Getis-Ord's index. Accordingly, for large spatial datasets, an approximate relation is as below:

$$G^*(r) - G(r) \approx \frac{\mathbf{u}^{\mathrm{T}}\mathbf{u}}{N_1}r^{-D_c}. \tag{32}$$

So far, the common spatial autocorrelation coefficients, including Moran's index, Geary's coefficient, and Getis-Ord's index, have been generalized to spatial autocorrelation functions. All these spatial autocorrelation functions have been associated with spatial correlation dimension. Thus, Based on the ideas from fractals, three types of spatial autocorrelation measurements have been integrated into the same logic framework of spatial analysis (Table 4).

The derivation results suggest that the spatial correlation dimension reflect both the spatial autocorrelation and spatial interaction. Moran's index is a spatial correlation coefficient,

**Table 4. The main mathematical relations between spatial correlation dimension and spatial autocorrelation statistics.**

| Statistic | Relation | Formula |
|---|---|---|
| **Moran's $I$** | Exact relation | $I^*(r) - \frac{I(r)}{1+N/M_0(r)} = \frac{N}{N_1} r^{-D_c}$ |
| | Approximation relation | $I^*(r) - I(r) \approx \frac{N}{N_1} r^{-D_c}$ |
| **Getis-Ord's $G$** | Exact relation | $G^*(r) - \frac{G(r)}{1+N/M_0(r)} = \frac{\mathbf{u}^{\mathrm{T}}\mathbf{u}}{N_1} r^{-D_c}$ |
| | Approximation relation | $G^*(r) - G(r) \approx \frac{\mathbf{u}^{\mathrm{T}}\mathbf{u}}{N_1} r^{-D_c}$ |
| **Geary's $C$** | Exact relation | $C_g(r) = \frac{n-1}{n}\left[\mathbf{o}^{\mathrm{T}}\mathbf{W}(r)\mathbf{z}^2 + \frac{N}{M_0(r)} - \frac{N_1 I^*(r)}{M_0(r)} r^{D_c}\right]$ |
| | Approximation relation | $C_g(r) \approx \mathbf{o}^{\mathrm{T}}\mathbf{W}(r)\mathbf{z}^2 - \frac{N_1 I^*(r)}{M_0(r)} r^{D_c}$ |

**Note**: In the equations of this table, $\mathbf{o}$ denotes ones vector $\mathbf{o} = [1, 1, \ldots, 1]^{\mathrm{T}}$.

Geary's coefficient is a spatial Durbin-Watson statistic, while Getis-Ord's index proved to be equivalent to the potential formula under certain conditions. Moran's index and Geary's coefficient reflect the extent and property of spatial autocorrelation, while Getis-Ord's index reflect both the spatial autocorrelation and spatial interaction. All these spatial statistics are associated with the spatial correlation dimension. In this sense, the spatial correlation dimension contain two aspects of geographical spatial information: spatial autocorrelation and spatial interaction. It is easy to calculate the spatial autocorrelation functions based on Geary's coefficient $C_g(r)$ and Getis-Ord's index $G(r)$, and the results correspond to Moran's function $I(r)$ (Table 5).

The ideas from correlation are important in the research on both city fractals and fractal cities. As indicated above, one of fractal dimension definitions is based on generalized correlation function. Spatial correlation can be divided into four types based on equation (9) (Fig 1). If $r$ is a constant, we will have a correlation based on fixed scale, which is used to define the common spatial autocorrelation coefficient; if $r$ depends on the size of geographical elements, we will have correlation based on characteristic scales; if $r$ is a variable but $i$ or $j$ is fixed to a certain element, we have a local scaling correlation, which can be used to define radial dimension of cities; if $r$ is a variable and $i$ and $j$ are not fixed to a certain element, we have a global scaling correlation, which can be used to define spatial correlation dimension derived above. The local correlation is termed one point correlation or central correlation, while the global correlation is termed point-point correlation or density-density correlation [23]. The former reflects the 1-dimensional correlation, while the latter reflect the 2-dimensional correlation. Spatial correlation is one of approaches to estimating fractal dimension of cities [48, 60, 61]. A number of interesting studies have been made to calculate fractal dimension of urban form, and the method can be combined with dilation method [62–66]. The spatial correlation can be integrated into the percolation analysis to model the complex evolution of urban growth [67–69]. The above results form a bridge between spatial correlation of urban patterns and spatial autocorrelation of geographical processes by means of the concepts from fractals and scaling.

Compared with the previous studies on spatial autocorrelation, this work is based on scaling law rather than characteristic scales. In literature, some studies involved the concept of spatial scaling, but do not delve into the essence of scaling laws [55]. Some scholars attempt to lessen the influence of spatial autocorrelation in the process of conducting scaling analysis of urban hierarchies [70]. The novelty of this paper lies in revealing the mathematical relationships between spatial autocorrelation functions and spatial correlation dimension. First, the strict mathematical relation between Moran's index set and spatial correlation dimension was derived. Thus, the common spatial analysis based on characteristic scale is associated with the spatial analysis based on scaling law. Second, the above-mentioned relation was generalized to

**Table 5. Datasets for spatial autocorrelation functions based on Geary's coefficient and Getis-Ord's index (Partial results).**

| Scale | 2000 (Fifth census data) | | | | 2010 (Sixth census data) | | | |
|---|---|---|---|---|---|---|---|---|
| | D+V | | N+V | | D+V | | N+V | |
| $r$ | Geary $C_g^*(r)$ | Getis $G^*(r)$ | Geary $C_g(r)$ | Getis $G(r)$ | Geary $C_g^*(r)$ | Getis $G^*(r)$ | Geary $C_g(r)$ | Getis $G(r)$ |
| 150 | 0.0770 | 0.0021 | 1.1934 | 0.0052 | 0.0931 | 0.0023 | 1.4432 | 0.0068 |
| 250 | 0.4366 | 0.0019 | 1.7027 | 0.0020 | 0.3687 | 0.0021 | 1.4379 | 0.0024 |
| 350 | 0.7660 | 0.0019 | 1.8767 | 0.0019 | 0.7144 | 0.0020 | 1.7502 | 0.0021 |
| 450 | 0.7343 | 0.0017 | 1.3607 | 0.0015 | 0.6769 | 0.0018 | 1.2542 | 0.0016 |
| 550 | 0.7619 | 0.0016 | 1.1565 | 0.0014 | 0.7835 | 0.0016 | 1.1892 | 0.0015 |
| 650 | 0.8146 | 0.0014 | 1.1338 | 0.0012 | 0.8068 | 0.0015 | 1.1230 | 0.0013 |
| 750 | 0.8517 | 0.0014 | 1.1038 | 0.0012 | 0.9123 | 0.0015 | 1.1822 | 0.0013 |
| 850 | 0.9366 | 0.0014 | 1.1835 | 0.0012 | 0.9996 | 0.0014 | 1.2631 | 0.0013 |
| 950 | 0.8701 | 0.0013 | 1.0703 | 0.0011 | 0.9148 | 0.0013 | 1.1254 | 0.0012 |
| 1050 | 0.9711 | 0.0013 | 1.1493 | 0.0012 | 1.0103 | 0.0013 | 1.1957 | 0.0012 |
| 1150 | 0.9265 | 0.0013 | 1.0757 | 0.0012 | 0.9705 | 0.0013 | 1.1268 | 0.0012 |
| 1250 | 0.9994 | 0.0014 | 1.1276 | 0.0013 | 1.0329 | 0.0014 | 1.1654 | 0.0013 |
| 1350 | 1.0589 | 0.0014 | 1.1743 | 0.0013 | 1.0789 | 0.0014 | 1.1965 | 0.0013 |
| 1450 | 1.0060 | 0.0013 | 1.1032 | 0.0013 | 1.0407 | 0.0013 | 1.1413 | 0.0013 |
| 1550 | 1.0299 | 0.0014 | 1.1221 | 0.0014 | 1.0574 | 0.0014 | 1.1520 | 0.0013 |
| 1650 | 1.0240 | 0.0014 | 1.1084 | 0.0013 | 1.0531 | 0.0014 | 1.1398 | 0.0013 |
| 1750 | 1.0118 | 0.0014 | 1.0916 | 0.0013 | 1.0367 | 0.0014 | 1.1184 | 0.0013 |
| 1850 | 0.9820 | 0.0013 | 1.0518 | 0.0013 | 1.0078 | 0.0013 | 1.0794 | 0.0013 |
| 1950 | 0.9536 | 0.0013 | 1.0162 | 0.0013 | 0.9684 | 0.0013 | 1.0319 | 0.0012 |
| 2050 | 0.9304 | 0.0013 | 0.9876 | 0.0013 | 0.9429 | 0.0013 | 1.0009 | 0.0013 |
| 2150 | 1.0066 | 0.0013 | 1.0630 | 0.0013 | 1.0163 | 0.0013 | 1.0732 | 0.0013 |
| 2250 | 1.0119 | 0.0013 | 1.0657 | 0.0013 | 1.0212 | 0.0013 | 1.0755 | 0.0013 |
| 2350 | 0.9789 | 0.0013 | 1.0277 | 0.0012 | 0.9937 | 0.0013 | 1.0432 | 0.0012 |
| 2450 | 1.0503 | 0.0013 | 1.1007 | 0.0013 | 1.0518 | 0.0013 | 1.1023 | 0.0013 |
| 2550 | 1.0254 | 0.0013 | 1.0720 | 0.0013 | 1.0275 | 0.0013 | 1.0742 | 0.0013 |
| 2650 | 1.0372 | 0.0013 | 1.0831 | 0.0012 | 1.0403 | 0.0013 | 1.0862 | 0.0013 |
| 2750 | 1.0434 | 0.0013 | 1.0886 | 0.0012 | 1.0477 | 0.0013 | 1.0930 | 0.0012 |
| 2850 | 1.0370 | 0.0013 | 1.0813 | 0.0012 | 1.0408 | 0.0013 | 1.0852 | 0.0012 |
| 2950 | 1.0271 | 0.0013 | 1.0704 | 0.0012 | 1.0304 | 0.0013 | 1.0738 | 0.0012 |
| 3050 | 1.0329 | 0.0013 | 1.0757 | 0.0012 | 1.0326 | 0.0013 | 1.0753 | 0.0012 |

**Note:** (1) The yardstick length $r$ represents measurement scales and displacement parameter of spatial correlation. (2) Difference scales $r$ lead to different Geary's coefficients $C$ and Getis-Ord's index $G$, which form Geary's function $C_g(r)$ and Getis-Ord's function $G(r)$. (3) D implies that diagonal elements are taken into account, N means that diagonal elements are removed, and V denotes variable mean values of spatial contiguity matrix elements.

Ceary's coefficient and Getis-Ord's index. This is helpful for developing a logic framework for spatial scaling analysis based on autocorrelation functions. Third, a complete case analysis was presented. This is useful for understanding spatial autocorrelation of Chinese cities. Fractal dimension suggests scaling law, and scaling invariance suggests nonlinear spatial autocorrelation rather than linear autocorrelation. The linear autocorrelation process is simple and can be described and analyzed using linear algebra and statistics. The nonlinear autocorrelation process is complex and requires the introduction of scaling analysis ideas and methods based on linear algebra and statistics to effectively describe and analyze it.

A good academic research result lies in its inspiration effect rather than its perfection. The shortcoming of this work lies in three respects. First, the spatial weight matrix is based on conventional concept, and special spatial separation factors have not been considered. The spatial autocorrelation function and corresponding correlation dimension rely heavily on the form, structure and types of spatial weight matrices. Generally speaking, weighting matrices for spatial autocorrelation analysis are constructed by means of the concept of spatial proximity or distance-decay functions. However, in some cases, two locations which are close geographically but separated by other factors such as lakes, rivers, mountains, and buildings may not be treated as near neighbors. A new finding is that weighting matrices based on combination of geographical proximity with covariate information may solve the problem, and *B*-statistics as a new spatial measure emerged as the times require [24]. Unfortunately, how to use the *B*-statistic including covariate information to construct spatial autocorrelation function and estimate spatial correlation dimension is an outstanding issue. Second, the size variable fails to be introduce to improve the definition spatial correlation dimension. Where cities are concerned, the fractal dimension of spatial correlation depends on the spatial distribution rather than size distribution of cities. In contrast, spatial autocorrelation functions based on Moran's index and Getis-Ord's index comprise both spatial contiguity measure and size measure. In the future, the size measure should be introduced into the definition of spatial correlation dimension as a weight form. Third, the empirical analysis for the spatial autocorrelation functions based on Geary's coefficient and Getis-Ord's index are not made for the time being. Although different spatial autocorrelation functions proved to be equivalent to one another, it is still necessary to make case studies to verify the theoretical inferences. However, limited to the space of a paper, the related empirical studies are not implemented. Moreover, the empirical analysis is only based on the observational data of Chinese cities. If we can obtain the spatial dataset of other countries, maybe we can make a comprehensive positive studies. Unfortunately, due to the limitation of observed data, the work remains to be done in the future.

## 5 Conclusions

Significant autocorrelation of a spatial sampling result suggests invalid mean values, which leads to uncertain probability structure. In this case, conventional statistical analysis cannot lead to reliable inference for complex systems. It is necessary to find new methods based on scaling law concept for spatial analysis. Spatial autocorrelation functions are actually a type of generalized correlation functions, and correlation functions are the base for defining generalized correlation dimension. Using correlation dimension, we can characterize multifractal scaling of complex systems such as cities. One of simple and important approach to constructing spatial autocorrelation functions based on spatial autocorrelation coefficients is to make use of the relative step function based on variable distance threshold. Thus, we can derive the spatial correlation dimension from the spatial autocorrelation functions. The main conclusions of this study can be reached as follows. *First, the spatial correlation dimension can be calculated by means of the relationships between the standard spatial autocorrelation function and the generalized spatial autocorrelation function*. The spatial autocorrelation coefficients are not enough to reflect the complex dynamics process of geographical evolution. Spatial autocorrelation functions can be employed to characterize the spatio-temporal dynamics of geographical systems, but the measurement procedure and quantitative description are complicated. Using spatial correlation dimension, we can condense sets of spatial parameters into a simple number, and thus it is easy to make spatial analyses of geographical processes. *Second, the spatial correlation dimension depends on spatial contiguity matrix rather than the size measure of geographical elements*. Changing size measure such as city population does not influence the

relationships between spatial autocorrelation functions and spatial correlation dimension. However, changing distances between geographical elements in a region leads to different relationships between Moran's functions and yardstick length and thus results in different spatial correlation dimension values. This suggests that the common spatial correlation dimension depends on spatial distribution patterns instead of size distribution patterns. Further, this suggests that spatial contiguity represent slow variable while size measure represents fast variable in geo-spatial analysis. *Third, the scaling ranges of spatial correlation dimension reflect the geographical scope of spatial autocorrelation and interaction.* In theory, the spatial correlation dimension is absolute, but in practice, the spatial correlation dimension is a relative measure and is always valid within certain range of measurement scales. By means of log-log plots, the scaling range can be approximately identified visually. The scaling range corresponds to the scope of positive autocorrelation reflected by the generalized spatial autocorrelation function based on Moran's index. This implies that the scaling range represents a quantitative criterion of spatial agglomeration of geographical distributions.

## Supporting information

**S1 File. Mathematical derivation process.** This file provides a detailed derivation process for the relationship between spatial correlation dimension and spatial autocorrelation function based on Moran's *I*.
(DOCX)

**S1 Dataset. SACF for 2000.** This file includes the observational data for the year 2000 and the related calculations based on this data. By studying this data table, readers can easily understand the data processing in the paper and its corresponding relationship with mathematical models derived in this work.
(XLSX)

**S2 Dataset. SACF for 2010.** This file includes the observational data for the year 2010 and the related calculations based on this data. This example further displays the data processing process of the case analysis for readers' reference.
(XLSX)

## Acknowledgments

I would like to thank the anonymous reviewer and Dr. Ejigu whose interesting and constructive comments were very helpful in improving the quality of this paper.

## Author Contributions

**Conceptualization:** Yanguang Chen.

**Data curation:** Yanguang Chen.

**Formal analysis:** Yanguang Chen.

**Funding acquisition:** Yanguang Chen.

**Investigation:** Yanguang Chen.

**Methodology:** Yanguang Chen.

**Project administration:** Yanguang Chen.

**Resources:** Yanguang Chen.

**Software:** Yanguang Chen.

**Supervision:** Yanguang Chen.

**Validation:** Yanguang Chen.

**Visualization:** Yanguang Chen.

**Writing – original draft:** Yanguang Chen.

**Writing – review & editing:** Yanguang Chen.

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
