## [Decision Letter · Decision Letter 0]

16 Feb 2024

PONE-D-23-11971Derivation of Correlation Dimension from Spatial Autocorrelation FunctionsPLOS ONE

Dear Dr. Chen,

Thank you for submitting your manuscript to PLOS ONE. After careful consideration, we feel that it has merit but does not fully meet PLOS ONE’s publication criteria as it currently stands. Therefore, we invite you to submit a revised version of the manuscript that addresses the points raised during the review process.

We look forward to receiving your revised manuscript.

Kind regards,

Bedilu Alamirie Ejigu, Ph.D

Academic Editor

PLOS ONE

Journal Requirements:

Additional Editor Comments:

Editor comment:

The paper's primary contribution lies in its ability to compute spatial correlation dimension using the standard spatial autocorrelation function. However, the manuscript lacks grammatical clarity and a coherent flow. In addition to editorial revisions, the authors should provide a clear description of how users can employ their method. It would be beneficial to include any R-function for implementation, along with brief instructions, as supplementary material.

It is unfortunate that many reviewers declined to review this manuscript, some even after agreeing to do so. To prevent unnecessary delays, the author must submit a revised version based on my assessment and one reviewer's comment. Since the submitted version lacked line numbers, it was difficult to locate the comments.

Specific comments.

Abstract:

- Rewrite it with subheading: Background, Method, Result and Conclusion

Introduction:

- Review https://doi.org/10.1016/j.spasta.2020.100454

Theoretical models

- On page 6 (last line), correct dij=0, is it to mean dij>r

- There are so many equations in the manuscript (above 45). Take many of the equation presented from page 7-10 to the appendix with additional details.

- In one paragraph clearly summarize the distinct features of your approach

Empirical analysis

- Ident (to the left) the computational steps outlined from P.10 to P.11.

- Table 3 summarized different statistics. What about the B-statistics (https://doi.org/10.1016/j.spasta.2020.100454)

- How the parameters in Eqn 35 obtained?

- Briefly summarize the results presented under Table 2.

Discussion

- Explain the added value of the proposed method over Moran’s I and B-statistic.

Reviewers' comments:

Reviewer's Responses to Questions

**Comments to the Author**

1. Is the manuscript technically sound, and do the data support the conclusions?

Reviewer #1: Yes

2. Has the statistical analysis been performed appropriately and rigorously? 

Reviewer #1: Yes

3. Have the authors made all data underlying the findings in their manuscript fully available?

Reviewer #1: Yes

4. Is the manuscript presented in an intelligible fashion and written in standard English?

Reviewer #1: Yes

5. Review Comments to the Author

Reviewer #1: This study reveals the inherent association of fractal patterns with spatial autocorrelation processes. The overall idea of the article is reasonable, and the technical approach is relatively mature. However, there are also some shortcomings. The author should revise the paper carefully according to the comments.

Specific Opinions:

1. The abstract of the article lacks data support. The results of the study are explained in detail in the abstract, but the presentation of specific experimental data is lacking. Authors are advised to revise the abstract and add a description of the data used.

2. Necessary citation information can enhance the reliability and accuracy of the article. In terms of literature citation, this article is mainly related to the older years. In order to reflect the research significance and value of this study, it is suggested that the authors should add citations of relevant articles in full-text journals in recent years.

3. The innovation point of the research is not sufficiently concise. The most outstanding innovation of this study is that it reveals the intrinsic relationship between fractal patterns and spatial autocorrelation processes. It is recommended to polish and revise the stiff parts of the creative point writing.

6. PLOS authors have the option to publish the peer review history of their article (what does this mean?). If published, this will include your full peer review and any attached files.

Reviewer #1: No

---

## [Author Response · Author response to Decision Letter 0]

8 Mar 2024

See the attached file entitle "Response to reviewers"

---

## [Editor Report · Decision Letter 1]

22 Apr 2024

Derivation of Correlation Dimension from Spatial Autocorrelation Functions

PONE-D-23-11971R1

Dear Dr. Chen,

We’re pleased to inform you that your manuscript has been judged scientifically suitable for publication and will be formally accepted for publication once it meets all outstanding technical requirements.

Kind regards,

Bedilu Alamirie Ejigu, Ph.D

Academic Editor

PLOS ONE
---

## [Editor Report · Acceptance letter]

21 May 2024

PONE-D-23-11971R1 

PLOS ONE

Dear Dr. Chen, 

I'm pleased to inform you that your manuscript has been deemed suitable for publication in PLOS ONE. Congratulations! Your manuscript is now being handed over to our production team.

Kind regards, 

on behalf of

Dr. Bedilu Alamirie Ejigu 

Academic Editor

PLOS ONE